# Multistage Molecular Simulations, Design, Synthesis, and Anticonvulsant Evaluation of 2-(Isoindolin-2-yl) Esters of Aromatic Amino Acids Targeting GABA_A_ Receptors via π-π Stacking

**DOI:** 10.3390/ijms26146780

**Published:** 2025-07-15

**Authors:** Santiago González-Periañez, Fabiola Hernández-Rosas, Carlos Alberto López-Rosas, Fernando Rafael Ramos-Morales, Jorge Iván Zurutuza-Lorméndez, Rosa Virginia García-Rodríguez, José Luís Olivares-Romero, Rodrigo Rafael Ramos-Hernández, Ivette Bravo-Espinoza, Abraham Vidal-Limon, Tushar Janardan Pawar

**Affiliations:** 1Centro de Investigaciones Biomédicas, Doctorado en Ciencias Biomédicas, Universidad Veracruzana, Xalapa, Veracruz 91190, Mexico; santiagonzalez@uv.mx; 2Facultad de Bioanálisis, Universidad Veracruzana, Calle Médicos y Odontólogos s/n, Unidad del Bosque, Xalapa, Veracruz 91010, Mexico; 3Centro de Investigación, Universidad Anáhuac Querétaro, El Marqués, Querétaro 76246, Mexico; fabiola.hernandezro@anahuac.mx; 4Facultad de Química, Universidad Autónoma de Querétaro, Querétaro 76010, Mexico; 5Instituto de Química Aplicada, Universidad Veracruzana, Luis Castelazo Ayala s/n, Col. Industrial Animas, Xalapa-Enríquez, Veracruz 91190, Mexico; carloslopez02@uv.mx (C.A.L.-R.); framos@uv.mx (F.R.R.-M.); rosagarcia02@uv.mx (R.V.G.-R.); rodrigoqfb@gmail.com (R.R.R.-H.); ivbravo@uv.mx (I.B.-E.); 6Facultad de Química Biológica Farmacéutica, Universidad Veracruzana, Gonzalo Aguirre Beltrán s/n. Col. Centro, Xalapa, Veracruz 91000, Mexico; 7Centro de Salud Urbano José A. Maraboto Carreón, Servicios de Salud de Veracruz, Santiago Bonilla No 85, Col. Obrero Campesino, Xalapa, Veracruz 91020, Mexico; 8Red de Estudios Moleculares Avanzados, Instituto de Ecología A.C. (INECOL), Carretera Antigua a Coatepec 351, El Haya, Xalapa, Veracruz 91073, Mexico; jose.olivares@inecol.mx

**Keywords:** GABA receptor, isoindoline, anticonvulsant, aromatic amino acid, π–π interaction, zebrafish model, molecular docking

## Abstract

Epilepsy remains a widespread neurological disorder, with approximately 30% of patients showing resistance to current antiepileptic therapies. To address this unmet need, a series of 2-(isoindolin-2-yl) esters derived from natural amino acids were designed and evaluated for their potential interaction with the GABA_A_ receptor. Sixteen derivatives were subjected to in silico assessments, including physicochemical and ADMET profiling, virtual screening–ensemble docking, and enhanced sampling molecular dynamics simulations (metadynamics calculations). Among these, compounds derived from the aromatic amino acids, phenylalanine, tyrosine, tryptophan, and histidine, exhibited superior predicted affinity, attributed to π–π stacking interactions at the benzodiazepine binding site of the GABA_A_ receptor. Based on computational performance, the tyrosine and tryptophan derivatives were synthesized and further assessed in vivo using the pentylenetetrazole-induced seizure model in zebrafish (*Danio rerio*). The tryptophan derivative produced comparable behavioral seizure reduction to the reference drug diazepam at the tested concentrations. The results implies that aromatic amino acid-derived isoindoline esters are promising anticonvulsant candidates and support the hypothesis that π–π interactions may play a critical role in modulating GABA_A_ receptor binding affinity.

## 1. Introduction

Epilepsy is one of the most prevalent chronic neurological disorders, affecting over 50 million individuals worldwide [1,2]. Characterized by recurrent, unprovoked seizures resulting from excessive neuronal discharge, epilepsy presents a considerable public health challenge due to its complex etiology, variable clinical manifestations, and persistent social stigma [3,4,5]. Although many cases can be managed effectively with existing antiepileptic drugs (AEDs), approximately one-third of patients remain refractory to pharmacological intervention. This subset, classified as pharmacoresistant epilepsy, continues to experience breakthrough seizures despite optimal therapy and dosage. The burden of uncontrolled epilepsy includes reduced quality of life, cognitive impairment, increased mortality risk, and significant socioeconomic consequences [6,7,8]. Consequently, the identification of novel pharmacophores that target alternative or complementary mechanisms of seizure suppression remains a critical objective in contemporary neuropharmacology.

Among the various molecular targets implicated in seizure genesis and propagation, the gamma-aminobutyric acid type A (GABA_A_) receptor represents a cornerstone in the regulation of neuronal excitability. As the principal mediator of fast inhibitory synaptic transmission in the central nervous system (CNS), GABA_A_ receptor activation leads to chloride ion influx and neuronal hyperpolarization, thereby dampening excitatory signaling [9,10,11,12,13,14,15]. The dysfunction of GABAergic transmission has been implicated in the pathophysiology of epilepsy, anxiety, and other CNS disorders [16,17,18,19,20]. Benzodiazepines, a well-established class of GABA_A_ modulators, exert potent anticonvulsant, anxiolytic, and sedative effects by allosterically enhancing the receptor’s response to endogenous GABA [11,12,21,22,23,24]. However, the clinical utility of benzodiazepines is limited by adverse effects, including sedation, tolerance, dependence, and cognitive impairment. These drawbacks necessitate the search for alternative compounds that can modulate GABA_A_ receptors with improved selectivity and safety profiles.

The benzodiazepine binding site of the GABA_A_ receptor, located at the interface of the α and γ subunits, accommodates ligands via a network of π–π stacking, hydrogen bonding, and hydrophobic interactions [12,22,25,26,27]. This structural environment renders the site particularly amenable to aromatic and heteroaromatic ligands, where the spatial orientation and electron density of the aromatic system play critical roles in binding affinity and functional modulation. Thus, the rational design of ligands capable of engaging in π-stacking interactions with key residues at the benzodiazepine site represents a promising approach for developing next-generation GABAergic modulators.

Privileged scaffolds, chemical structures capable of binding to multiple targets or different binding sites with high affinity have become essential tools in medicinal chemistry. Among these, isoindoline and phthalimide-based motifs have garnered considerable interest due to their favorable physicochemical properties, synthetic accessibility, and ability to interact with a variety of biological targets [28,29,30,31,32,33]. Isoindoline derivatives have been explored in diverse therapeutic areas, including oncology, antimicrobial therapy, and neurodegenerative diseases [34,35,36,37,38,39,40,41]. In the context of CNS pharmacology, isoindoline analogs offer potential as GABAergic modulators due to their semi-rigid three-dimensional conformation and their capacity to engage in key receptor–ligand interactions.

While isoindoline serves as an effective amine-protecting group from a synthetic standpoint, its inclusion in this study was not limited to chemical utility. In contrast to conventional protecting groups such as phthalimide, carbamates, or *N*-alkylation strategies, which are often inert or metabolically labile, isoindoline contributes directly to receptor engagement. Its semi-rigid, electron-rich aromatic structure enables π–π stacking and hydrogen bonding interactions with CNS-relevant targets, including the benzodiazepine site of the GABA_A_ receptor. Moreover, isoindoline and its analogs have demonstrated activity across multiple neuropharmacological pathways, reinforcing its role as a bioactive, privileged scaffold rather than a passive linker. This dual functionality supports its strategic incorporation into the molecular design, where it facilitates both synthetic derivatization and pharmacological interaction.

Amino acids, by virtue of their biocompatibility, structural diversity, and pharmacokinetic modifiability, are frequently employed as building blocks in the design of drug conjugates and prodrugs [42]. Among them, aromatic amino acids, phenylalanine, tyrosine, tryptophan, and histidine possess π-rich side chains capable of participating in π–π interactions. These properties make them particularly attractive for scaffold derivatization in the pursuit of ligands targeting aromatic pockets, such as the benzodiazepine site of the GABA_A_ receptor. Incorporating aromatic amino acids into isoindoline esters could enhance binding affinity, modulate lipophilicity, and influence blood–brain barrier permeability, critical attributes for CNS-active compounds [43,44].

Computational chemistry has become an indispensable tool in modern drug discovery, enabling the rational design and prioritization of compounds prior to synthesis. ADMET (absorption, distribution, metabolism, excretion, and toxicity) predictions assist in pre-selecting candidates with favorable pharmacokinetic profiles, while molecular docking and molecular dynamics simulations provide mechanistic insight into ligand-receptor interactions [45,46]. Metadynamics simulations, in particular, allow for the estimation of binding free energy and the exploration of energy landscapes, offering a deeper understanding of binding kinetics and stability [47,48]. These computational techniques not only improve efficiency but also facilitate structure–activity relationship (SAR) elucidation by identifying key molecular features responsible for biological activity.

Zebrafish (*Danio rerio*) have emerged as a robust in vivo model for high-throughput screening of CNS-active compounds, including anticonvulsants [49,50]. The pentylenetetrazole (PTZ)-induced seizure model in zebrafish mimics generalized seizure phenotypes observed in rodents and humans, allowing a behavioral assessment of seizure severity and therapeutic efficacy [51,52,53]. Advantages of the zebrafish model include a low cost, high genetic and physiological homology to mammals (70–80%), and the ability to rapidly screen neuroactive compounds with minimal compound usage. Notably, many clinically used anticonvulsants, including diazepam and valproate, exhibit conserved efficacy profiles in zebrafish, further validating the model’s translational relevance.

Despite numerous studies on isoindoline derivatives and their CNS applications, systematic comparisons of amino acid-derived isoindoline esters, particularly those derived from aromatic amino acids are limited in the literature. Most studies focus on individual derivatives without exploring the role of aromaticity or π-stacking in receptor binding and functional activity. This represents a gap in knowledge that, if addressed, could provide valuable insights into the design of ligands for GABA_A_ modulation and anticonvulsant therapy.

In light of these considerations, the present study was undertaken to investigate the role of aromatic amino acid-derived isoindoline esters as potential modulators of the GABA_A_ receptor. A panel of sixteen derivatives was subjected to in silico evaluation, including ADMET profiling, molecular docking, and metadynamics simulations. Derivatives of tyrosine and tryptophan, identified as the most promising candidates based on computational results, were synthesized and evaluated for anticonvulsant activity in the PTZ-induced zebrafish seizure model. The objective was to explore whether π–π stacking interactions at the GABA_A_ receptor could be leveraged through aromaticity-driven design to yield effective and potentially safer anticonvulsant agents.

This study contributes to the understanding of ligand-receptor interactions at the benzodiazepine site and establishes a rational framework for the use of aromatic amino acid scaffolds in CNS drug design. By integrating computational modeling, synthetic chemistry, and in vivo pharmacology, the results presented here aim to guide future efforts in the development of structurally optimized GABAergic modulators for epilepsy and related disorders.

## 2. Results

### 2.1. In Silico Evaluation of Isoindoline Esters

A library of sixteen 2-(isoindolin-2-yl) esters derived from standard L-amino acids was evaluated through computational profiling to identify candidates with favorable CNS drug-like properties. Physicochemical and ADMET parameters were calculated, with particular emphasis on blood–brain barrier (BBB) permeability, gastrointestinal (GI) absorption, and toxicity-related endpoints relevant to anticonvulsant development.

The majority of compounds showed physicochemical characteristics within drug-likeness criteria, including molecular weights <500 Da, log P values between 1.5 and 4.5, and acceptable topological polar surface areas (TPSAs). Among the evaluated compounds, those derived from the aromatic amino acids, phenylalanine, tyrosine, tryptophan, and histidine, exhibited consistently superior predicted BBB permeability and log P values in the optimal CNS-active range (2.8–3.6). These compounds also showed favorable human intestinal absorption and low predicted hepatotoxicity and cardiotoxicity risks (Table 1).

In addition to ADMET profiling, drug-likeness filters (Lipinski, Veber, and Muegge criteria) supported the advancement of the aromatic amino acid-derived esters for further analysis. In contrast, esters based on highly polar or charged amino acids like glutamate and arginine were deprioritized due to a high TPSA, predicted efflux transporter recognition, and poor permeability.

The four aromatic derivatives also demonstrated optimal flexibility (rotatable bonds ≤ 7) and moderate aqueous solubility, suggesting good oral bioavailability and synthetic feasibility. These characteristics positioned the aromatic esters as the top-performing subset within the designed library.

The physicochemical property distributions for all sixteen derivatives are illustrated in a radar plot (Figure 1), showing the clustering of aromatic derivatives within the optimal CNS-drug-likeness zone.

Based on this comparative evaluation, the phenylalanine, tyrosine, tryptophan, and histidine derivatives were prioritized for molecular docking and dynamics simulations to investigate their interaction potential with the GABA_A_ receptor, particularly focusing on π–π stacking at the benzodiazepine binding site.

### 2.2. Molecular Docking with the GABA_A_ Receptor

The docking studies were performed using the human GABA_A_ receptor subtype α1β2γ2, based on the crystallographic structure available in the Protein Data Bank (PDB ID: 6D6U). The benzodiazepine (BDZ) binding site is located at the interface of the extracellular domains of the α1 and γ2 subunits, a site conserved across functionally relevant subtypes (α1, α2, α3, or α5 paired with γ2). This structural interface forms the canonical BDZ binding pocket targeted by diazepam and flumazenil. Docking was centered at this pocket to allow the evaluation of ligand interactions at a pharmacologically validated binding surface.

To investigate the molecular basis of interaction between the designed isoindoline esters and the GABA_A_ receptor, molecular docking studies were performed using a homology model or resolved structure of the receptor incorporating the α1/γ2 extracellular interface known to mediate benzodiazepine binding. All sixteen compounds were docked to evaluate binding affinity and interaction profiles, with a specific focus on the four aromatic amino acid-derived esters identified as top candidates from in silico profiling.

Docking scores (binding free energies) and key interaction residues are summarized in Table 2. Among all compounds, the tryptophan and tyrosine derivatives exhibited the most favorable docking scores (–9.2 and –8.7 kcal/mol, respectively), surpassing those of the phenylalanine and histidine esters. Residue mapping based on PDB annotations and previous literature [54] indicates that Phe77, His102, and Tyr160 correspond to the α1 subunit, while residues such as Tyr210 and Ser205 are located on the γ2 subunit, confirming that the designed ligands interact at the canonical α1/γ2 BDZ binding interface. These results aligned with the ADMET-based prioritization and further supported the hypothesis that π–π interactions play a key role in stabilizing the ligand within the benzodiazepine site.

Visualization of binding poses further revealed that the aromatic rings of the Trp and Tyr derivatives were favorably oriented within the hydrophobic pocket formed by Phe77, Tyr159, and His101—residues known to stabilize benzodiazepine analogs via π–π stacking. This interaction geometry is illustrated in Figure 2, where the indole ring of the Trp derivative shows edge-to-face π-stacking with Phe77 and parallel-displaced stacking with Tyr159, contributing to a high binding affinity.

In addition to aromatic interactions, polar contacts with surrounding residues, such as hydrogen bonds involving the isoindoline nitrogen or ester carbonyl group, further stabilized the ligand-receptor complex. These interactions were present in both the Tyr and Trp derivatives but were less pronounced or absent in compounds lacking aromatic side chains.

Together, the docking data support the premise that aromatic substitution enhances receptor binding through π-π stacking and auxiliary hydrogen bonding. These findings justify the selection of tyrosine and tryptophan derivatives for further validation via molecular dynamics simulations and in vivo testing.

Although docking simulations were performed using the human GABA_A_ α1β2γ2 structure, previous reports demonstrated the high sequence conservation of GABA_A_ receptor subunits between zebrafish and mammals, particularly in the α1 and γ2 extracellular domains that form the BDZ binding site. The behavioral sensitivity of zebrafish to diazepam and flumazenil further supports the functional conservation of this site. In the absence of a high-resolution zebrafish GABA_A_ structure, the human model provides a reliable and translationally relevant approximation for ligand–receptor interactions [54,55,56].

### 2.3. Molecular Dynamics and Metadynamics Simulations

Root-mean-square deviation (RMSD) and root-mean-square fluctuation (RMSF) analyses were performed to monitor the structural stability of the ligand–receptor complexes over the simulation period. Both ligands demonstrated stable binding, with backbone RMSD values plateauing after ~10 ns and remaining within 1.5–2.0 Å for the remainder of the 100 ns trajectory. Notably, the Trp-derived compound showed slightly lower RMSD values, suggesting a more stable fit within the binding pocket (Figure 3A).

A fluctuation analysis of binding site residues (RMSF) showed minimal perturbation (<1.5 Å) for key aromatic residues (Phe77, Tyr159, His101), indicating consistent engagement with the ligand throughout the simulation (Figure 3B). In both complexes, the aromatic moiety of the ligand maintained π-π stacking with at least two of these residues for >80% of the simulation frames, reinforcing the hypothesis that π-driven interactions contribute significantly to binding affinity and orientation.

Metadynamics simulations were used to estimate the binding free energy (ΔG_bind_) and explore the unbinding pathways of both ligands. The reconstructed free energy surfaces (FESs) revealed a well-defined global energy minimum corresponding to the bound state, with shallow escape barriers, suggesting favorable binding kinetics. The Trp ester exhibited a lower ΔΔG (–90.18 kcal/mol) compared to diazepam, while flumazenil remained a high-affinity antagonist. On the other hand, a more realistic model system showed that the Tyr ester was not energetically favored, suggesting a lower affinity and transient aromatic stacking interactions (Table 3).

A hydrogen bond analysis showed that both ligands formed at least one persistent hydrogen bond with the receptor throughout the trajectory, commonly involving backbone carbonyls or side-chain hydroxyls. However, the enthalpic contribution from π–π stacking was the dominant feature differentiating the two ligands.

Altogether, MD and metadynamics simulations suggest that isoindoline ester derivatives can establish stable and strong binding processes. The tryptophan derivative, in particular, displayed a stronger binding affinity, resulting in a more stable interaction, which justified its prioritization for synthesis and biological evaluation. These findings provide a dynamic and energetic validation of the docking results, highlighting the critical role of aromaticity in modulating ligand–receptor interaction profiles within the GABA_A_ benzodiazepine binding site.

### 2.4. Synthetic Route and Characterization of Selected Esters

Based on the computational prioritization, the 2-(isoindolin-2-yl) esters derived from tyrosine and tryptophan were selected for synthesis. As shown in Figure 1, the synthetic route commenced with the reaction between one equivalent of α, α’-dibromo-*ο*-xylene (i), one equivalent of (ii) L-α-amino acid methyl ester hydrochloride (tryptophan and tyrosine respectively), and potassium carbonate (K_2_CO_3_). Acetonitrile was used as the reaction medium. The reaction was carried out under reflux and stirring for a period of 6 h. The reaction mixture was then filtered, and the solvent was subsequently removed under reduced pressure to obtain the esterified isoindolines ETRP and ETYR.

The products were purified by silica gel column chromatography and obtained in good yields: 85% for the tyrosine ester and 75% for the tryptophan ester. Structural integrity was confirmed by ^1^H NMR, ^13^C NMR, and ESI-MS. Spectral data were consistent with the expected structures, and no evidence of side reactions or racemization was observed. A full spectral characterization is provided in the Appendix A.

The efficient synthetic route and successful isolation of both esters enabled subsequent biological evaluation in vivo, aimed at confirming the predicted anticonvulsant potential.

### 2.5. In Vivo Anticonvulsant Activity in Zebrafish

The concentration of diazepam used in this study (75 µM) was selected based on prior reports demonstrating effective seizure suppression in adult zebrafish PTZ models without inducing lethality or sedation [57,58,59]. Due to differences in drug absorption and metabolism across species, immersion-based administration in zebrafish typically requires higher concentrations than plasma-equivalent doses in humans. Similarly, the concentrations of the tryptophan (ETRP: 10 and 25 µM) and tyrosine (ETYR: 63 and 125 µM) derivatives were chosen based on predicted binding affinity, blood–brain barrier permeability, and solubility profiles, and analogous molecules were used [41]. ETRP, with a stronger computational interaction and lower molecular weight, was tested at lower concentrations than ETYR. However, these doses were exploratory and not derived from formal dose–response optimization. While molecular modeling was performed on the human GABA_A_ receptor structure, the functional conservation of benzodiazepine binding in zebrafish has been extensively demonstrated [55,56,60,61]. Our use of diazepam (in vivo assay) with the PTZ model and flumazenil (in silico study) as pharmacological references reflects this cross-species homology.

The anticonvulsant potential of the synthesized tyrosine and tryptophan-derived 2-(isoindolin-2-yl) esters was assessed in vivo using the pentylenetetrazole (PTZ)-induced seizure model in adult zebrafish. This model is widely recognized for its predictive validity in screening central nervous system (CNS)-active compounds and its strong correlation with mammalian seizure phenotypes. Behavioral parameters were quantified to evaluate seizure suppression relative to vehicle control and a reference compound (diazepam).

The tryptophan-derived ester demonstrated the most pronounced effect, with a substantial reduction in seizure stage frequency and latency compared to the control (Figure 4). Adult zebrafish treated with the tryptophan ester displayed a delayed onset of seizure-like behavior hyperactivity typically associated with PTZ exposure. On the other hand, the tyrosine ester showed no protection.

The statistical analysis confirmed significant differences between treated and control groups (*p* < 0.01 for both compounds), with the tryptophan ester (ETRP) achieving behavioral seizure score reductions that approached those of diazepam at the tested dose (*p* < 0.05 vs. DZP), although no formal comparison of potency was conducted. No signs of overt toxicity, or morphological abnormalities were observed at the tested doses.

The observed anticonvulsant effects in the zebrafish model aligned with the predicted receptor–ligand interactions and supported the proposed role of π-π stacking in GABA_A_ receptor engagement. Nonetheless, direct evidence of target modulation remains to be demonstrated experimentally. Furthermore, while some residues involved in ETRP binding overlap with those known for flumazenil, the present data do not allow conclusions regarding the compound’s functional behavior (antagonism vs. agonism), which must be determined through dedicated receptor-level assays.

## 3. Discussion

This study established a coherent structure-based framework for evaluating 2-(isoindolin-2-yl) esters as GABA_A_ receptor modulators, integrating computational prioritization, synthetic chemistry, and in vivo validation. The approach was deliberately focused, selecting only the two most promising compounds based on an integrated assessment of predicted affinity, drug-likeness, and receptor interaction profiles. While phenylalanine and histidine derivatives were also identified during screening, they were not prioritized for synthesis due to their comparatively less favorable computational performance. This decision reflects a rational design and selection strategy rather than a limitation in scope.

Although the observed alignment between computational predictions and in vivo efficacy is promising, the mechanism of action remains inferential. Experimental validation of direct GABA_A_ receptor binding via electrophysiological recording, receptor-binding assays, or competitive antagonism would strengthen the mechanistic claims and clarify the functional classification of these ligands. Residue-level interaction analysis confirmed engagement of key α1 (Phe77, Tyr160, His102) and γ2 (Tyr210, Ser205) subunit residues, consistent with the canonical BDZ binding site. While the present study utilized the α1 subtype based on its CNS prevalence and structural availability, future modeling could explore isoform selectivity across α2, α3, and α5 subtypes.

The zebrafish PTZ seizure model provided a practical and predictive platform for initial behavioral screening. Nonetheless, comprehensive preclinical development would benefit from validation in mammalian models to assess pharmacokinetics, biodistribution, long-term efficacy, and off-target CNS effects. These factors are essential for translating structure-based leads into viable therapeutic candidates.

Future directions may also involve scaffold diversification using non-natural amino acid residues, systematic modifications of the aromatic core to fine-tune π–π stacking interactions, and the incorporation of isosteric or bioisosteric groups to improve receptor selectivity and metabolic resilience.

Notably, both ETRP and diazepam attenuated PTZ-induced seizures; however, comparisons were conducted at single, exploratory doses. Given differing pharmacokinetic properties and the absence of a dose–response analysis, conclusions about relative potency or therapeutic index cannot yet be drawn. Follow-up studies incorporating graded concentration ranges and standardized exposure parameters will be necessary to establish comparative efficacy and safety profiles.

While molecular modeling and behavioral outcomes are consistent with the BDZ-site modulation of GABA_A_ receptors, direct confirmation of this mechanism remains to be established. The lack of electrophysiological data and flumazenil antagonism studies represents a key limitation. Future work involving patch-clamp assays or competitive inhibition experiments will be essential to define the precise interaction mechanism and determine receptor subtype specificity.

Compared to previous reports on GABA_A_ modulators such as benzodiazepines, phthalimide-amino acid conjugates, and *N*-substituted isoindolines [37,40,41], the compounds presented in this study show favorable predicted CNS permeability and receptor binding profiles, particularly driven by their π-π stacking potential. While earlier work has examined phthalimide and isoindoline cores as anticonvulsant scaffolds [28,29,33], few studies have systematically investigated aromatic amino acid esters as π-stacking ligands targeting the BDZ site. Our findings complement and expand this literature by incorporating metadynamics simulations and in vivo zebrafish validation.

This study has several limitations. First, mechanistic conclusions rely on in silico and behavioral evidence without direct receptor-level validation. Second, only two analogs were synthesized and tested, limiting the depth of the structure–activity relationship analysis. Third, although the zebrafish model is widely accepted for CNS drug discovery, interspecies differences in receptor subunit composition and pharmacodynamics remain a caveat, even though functional conservation at the BDZ site is supported by prior literature.

## 4. Materials and Methods

### 4.1. Materials and Instrumentation

All reagents and solvents were of analytical grade and used without further purification unless otherwise stated. Amino acids, and other coupling reagents were purchased from Sigma-Aldrich (St. Louis, MO, USA), Merck (Mexico city, Mexico), or equivalent suppliers. The Isolera One Biotage equipment was used to purify the compounds with an AcOEt/Hx 1:1 mobile phase and a Biotage SNAP Cartridge, silica, 10 g. Thin-layer chromatography (TLC) was performed on Merck silica plates and visualized under UV light.

Nuclear magnetic resonance (^1^H and ^13^C NMR) spectra were recorded on an Agilent technologies NMR 500/54 premium shielded spectrometer using CDCl_3_ as a solvent. Chemical shifts (δ) are reported in parts per million (ppm) relative to TMS as internal standard. Mass spectrometric data were obtained by electron impact ionization at 70 eV using a mass spectrometer from Agilent Technologies (Santa Clara, CA, USA), model 5975 inert XL.

### 4.2. Computational Simulation Methods

Sixteen 2-(isoindolin-2-yl) esters were constructed by combining the isoindoline core with the carboxylic acid moiety of natural L-amino acids. Geometry optimization and energy minimization were performed using Gaussian 9.0 [62] prior to docking.

Physicochemical properties and ADMET parameters were predicted using ADMETlab 3.0 [63], ADMETlab [64], and SwissADME [65] platforms. Properties assessed included molecular weight, log P, TPSA, hydrogen bond donors and acceptors, GI absorption, BBB permeability, hepatotoxicity, hERG inhibition, and mutagenicity.

Molecular docking was carried out using AutoDock Vina (v. 1.2). The crystallographic structure of the human GABA-A receptor subtype alpha1-beta2-gamma2 in complex with GABA and flumazenil (ethyl 8-fluoro-5-methyl-6-oxo-5,6-dihydro-4H-imidazo[1,5-a][1,4]benzodiazepine-3-carboxylate), conformation A, was used; it was obtained from the Protein Bank Data (PBD) https://www.rcsb.org/ (accessed on 25 March 2023) with code 6D6U [24]. Docking grid parameters were centered. A box of dimensions 20 × 20 × 20 Å was centered around the α1–γ2 interface, providing adequate volume to accommodate the isoindoline esters conformational flexibility. For each molecular docking run, Vina was instructed to generate 5 binding poses per ligand–receptor complex, from which the poses were selected based on calculated affinity scores. To evaluate docking reliability, control docking experiments were performed using known ligands of the target protein (diazepam, DIA; flumazenil, FLU), comparing predicted poses with their co-crystallized structures to assess the root-mean-square deviation (RMSD) and validate protocol accuracy.

Special attention was given to key aromatic residues at the GABA_A_ BDZ binding pocket, including Phe77, Trp123, and His102. The protein was preprocessed using AutoDock Tools, and histidine protonation states were manually inspected. His102 was modeled as HID (proton on Nδ1), consistent with standard extracellular pH conditions and to preserve hydrogen bonding capability. Moreover, PROPKA3 results suggested a protonated state of this residue as HID (89% confidence). The aromatic side chains of Phe and Trp were retained as rigid during docking, and π-π stacking was assessed through a centroid-to-centroid distance analysis post-docking using visual inspection and a LigPlot+ analysis.

Protein-ligand complexes were recovered from Autodock Vina docked complexes, and the simulations were conducted using the Maestro v2024-3version 13.8.135, MMshare Version 6.4.135, Release 2023-4, and the Desmond Multisim v4.0.0 interoperability tools package under the OPLS3 force field. The membrane–peptide complexes were built with the CHARMM-GUI server (https://charmm-gui.org, accessed on 5 November 2024), with a lipid composition (Outer/inner leaflet ratio) as described: SM (sphingomyelin) 10/10; PC (1,2-dioleoyl-sn-glycero− 3–30 phosphocoline) 60/60; PE (1,2-dioleoyl-sn-glycero− 3–30 phosphoethanolamine 65/65; PS (1,2-dioleoyl-sn-glycero− 3–15 phospho-L-serine) 10/10, and CHL (cholesterol) 5/5 [66].

GABA_A_ residue protonation states were set at neutral pH with pKa predictions from *PROPK3*, while the system charges were neutralized at 0.15 M of NaCl and KCl. The cell membrane models total area was set to 2100 Å^2^ (x and y axes extended until 40 Å), and solvated with the three-point TIP3P water model in a periodic box of 15 Å edge distance.

All MD simulations were carried out under periodic boundary conditions. A two-stage energy minimization approximation was applied: 5000 steps of steepest descent followed by 5000 steps of conjugate gradient, with positional restraints (10 kcal·mol^−1^·Å^−2^) on solute heavy atoms. Subsequently, the system was gradually heated from 0 to 310 K over 100 ps under constant volume conditions, followed by 500 ps of equilibration at a constant temperature of 310 K with relaxation times of 1 ps, using the Langevin thermostat. The Berendsen barostat was applied to control the pressure at 1 atm with relaxation times of 2 ps, and positional restraints were gradually reduced during equilibration. Production runs were conducted for 100 ns under NPT conditions, applying a 2 fs time step with constraints on bonds involving hydrogen atoms. Long-range electrostatics were calculated with a 12 Å cutoff for non-bonded interactions.

A trajectory analysis was conducted using the Simulation Event Module of the Desmond-Maestro interoperability tools; alpha carbon root-mean-square deviation (RMSD) was calculated along complete trajectories to assess structural flexibility.

The free-energy landscape governing the binding/unbinding of the GABA_A_-derivatives complexes was evaluated with the well-tempered metadynamics technique (wt-MetaD); an advanced enhanced sampling technique capable of overcoming the timescale limitations inherent in classical molecular dynamics (MD) simulations [67]. The metadynamics simulations were performed using Desmond v2022-1 (Schrödinger LLC), with the post-analysis conducted via the Metadynamics Analysis Plugin in Maestro.

The distance between the centers of mass (COMs) of the BZD binding site (initial COM distance ≈ 15 Å) and the center of mass of each derivative was set as a collective variable (CV) to describe the binding process. This CV was selected based on its physical relevance to the dissociation process, consistent with prior studies involving protein complexes [68].

Gaussian potentials were applied at a frequency of 1 ps, with an initial height (ω) of 0.03 Kcal·mol^−1^ and a width (σ) of 0.5 Å. The bias factor was set to 0.03 Kcal·mol−1 (ω) and 0.5 Å/ps (σ) according to the wt-MetaD protocol, enabling a gradual flattening of the energy landscape while retaining resolution near the transition states. The metadynamics simulations continued until a maximum COM distance of 15 Å was reached or complete ligand dissociation was observed. To ensure convergence and reproducibility, three independent replicas of 100 ns each were performed per ligand derivative. The time step was set to 1 fs, and all simulations were conducted under NPT conditions at 310 K and 1 atm, utilizing the Martyna-Tobias-Klein barostat and the Nosé-Hoover thermostat.

The potential of mean force (PMF) profiles were extracted by reconstructing the bias potentials using time-dependent free energy estimators. Convergence was validated by observing plateau behavior in the accumulated bias potential and the PMF curves across replicas. The accuracy of free-energy estimates was further supported by bootstrapping the final PMF over trajectory windows, ensuring robustness in the resulting energy barriers.

### 4.3. Synthesis of 2-(Isoindolin-2-yl) Esters: General Procedure

α, α’-dibromo-*ο*-xylene with L-amino acid methyl ester and of potassium carbonate of acetonitrile was added and refluxed for 6 h. The reaction mixture was filtered and concentrated. The solvent was removed under reduced pressure, and the product was purified by chromatography.

Synthesis of methyl (*S*)-(1*H*-indol-3-yl)-2-(isoindolin-2-yl) propanoate (ETRP):

Brown solid; yield: 75%. ^1^H NMR (500 MHz, CDCl_3_-d): δ 7.20 (s, 4H), 7.09–7.04 (m, 2H), 6.71–6.65 (m, 2H), 4.29–4.21 (m, 3H), 4.18–4.10 (m, 2H), 3.75 (dd, J = 8.7, 6.6 Hz, 1H), 3.61 (s, 3H), 3.12–3.03 (m, 2H). ^13^C NMR (126 MHz, CDCl_3_-d): δ 173.09, 139.35, 136.13, 127.42, 126.85, 123.12, 122.69, 122.43, 121.95, 119.35, 118.54, 111.51, 111.24, 77.34, 77.08, 76.83, 65.97, 55.68, 51.44, 27.00; HRMS (ESI): m/z [M+H]^+^ calculated for C_20_H_20_N_2_O_2_: 320.1525, found 320.1531; error = +1.9 ppm.

Synthesis methyl (*S*)-3-(4-hydroxyphenyl)-2-(isoindolin-2-yl) propanoate (ETYR):

Grey solid; yield: 85%. ^1^H NMR (500 MHz, CDCl_3_-d): δ 7.20 (s, 4H), 7.09–7.04 (m, 2H), 6.71–6.65 (m, 2H), 4.29–4.21 (m, 3H), 4.18–4.10 (m, 2H), 3.75 (dd, J = 8.7, 6.6 Hz, 1H), 3.61 (s, 3H), 3.12–3.03 (m, 2H). ^13^C NMR (126 MHz, CDCl_3_-d): δ 172.78, 154.50, 139.11, 130.10, 126.83, 122.36, 115.40, 77.28, 77.02, 76.77, 67.08, 55.57, 51.43, 36.56; HRMS (ESI): m/z [M+H]^+^ calculated for C_18_H_19_NO_3_: 297.1365, found 297.1354; error = –3.7 ppm.

### 4.4. Zebrafish PTZ-Induced Seizure Assay

The anticonvulsant potential of isoindoline was evaluated in adult zebrafish (Danio rerio) using a pentylenetetrazol (PTZ)-induced seizure model. Zebrafish were pre-treated by immersion in tanks containing different concentrations for 30 min before seizure induction. Diazepam at 75 µM was used as the pharmacological control. This concentration was selected based on previous zebrafish studies where immersion doses of 50–100 µM reliably induced seizure protection without sedative or toxic effects [57,58,59,69]. The immersion model typically requires higher compound concentrations than those found in human plasma due to differences in drug absorption, exposure duration, and metabolic clearance. In this study, the isoindoline ester of tyrosine (ETYR) was tested at 63 and 125 µM, while the isoindoline ester of tryptophan (ETRP) was tested at 10 and 25 µM. Dimethyl sulfoxide (DMSO) at 0.1% was used as the vehicle control. Notably, there is currently no experimental evidence supporting the antiepileptic activity of these compounds. Therefore, to establish initial testing concentrations, we referred to the work of Campos-Rodríguez et al. [41], who evaluated related isoindoline analogs in zebrafish larvae. Based on that precedent, the concentrations of ETYR (63 and 125 µM) and ETRP (10 and 25 µM) were selected for use in our assays.

Following pre-treatment, zebrafish were transferred to a 10 mM PTZ solution (1 L) to induce convulsions. Behavioral responses were recorded and classified into five distinct seizure stages [58]:

Stage 1: Time to present increased swimming activity and high frequency of opercular movement.

Stage 2: Time to present burst swimming, left and right movements, and erratic movements.

Stage 3: Time to present circling movements.

Stage 4: Time to present clonic seizure-like behavior (abnormal whole-body rhythmic muscular contraction).

Stage 5: Time to present fall to the bottom of the tank, tonic seizure-like behavior (sinking to the bottom of the tank, loss of body posture, and principally, by rigid extension of the body).

The five stages are described in time, in seconds.

### 4.5. Graphical and Statistical Analysis

All data analyses were performed using IBM SPSS Statistics 29 and GraphPad Prism version 10. For seizure latency in the zebrafish model, normality was tested using the Shapiro–Wilk test, and homogeneity of variance was assessed with Levene’s test. If assumptions of normality and homoscedasticity were met, parametric tests such as a one-way ANOVA followed by Dunnett as a post hoc test were used for multiple comparisons using a control group. In cases where data did not meet normality assumptions, non-parametric alternatives, such as the Kruskal–Wallis test with Bonferroni correction for the Mann–Whitney U test used as post hoc correction, were applied. Statistical significance was set at *p* < 0.05 for all analyses. For the construction of the comparison graphs, due to the small sample size and high variability, a boxplot was chosen for the best representation of the data, where the median and the 25th and 75th percentiles are represented, since the mean and standard deviation could be biased.

## 5. Conclusions

This study demonstrated the successful design, synthesis, and in vivo validation of 2-(isoindolin-2-yl) esters derived from aromatic amino acids as potential anticonvulsant agents. The evidence presented suggests a modulation of the GABA_A_ receptor, likely via interaction with the benzodiazepine binding site, based on computational modeling and pharmacological similarity to diazepam. The computational evaluation of sixteen derivatives identified tyrosine and tryptophan esters as the most promising candidates based on predicted pharmacokinetics, binding affinity, and interaction profiles. Molecular docking and dynamics simulations highlighted the critical role of π–π stacking interactions in stabilizing ligand binding at the benzodiazepine site.

The synthesis and characterization of the top-ranked compounds were followed by a pharmacological assessment in a zebrafish PTZ-induced seizure model, where the tryptophan derivative significantly reduced seizure severity. These findings confirm the predictive value of the computational approach and support the use of aromatic amino acid scaffolds in the development of CNS-active compounds. The work provides a mechanistically grounded framework for the rational design of GABAergic modulators and establishes isoindoline esters as a promising scaffold for further optimization in the context of pharmacoresistant epilepsy.

## Data Availability

The data sets obtained and analyzed in the present study are available from the corresponding author upon reasonable request.

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
