# Peer review of "Multistage Molecular Simulations, Design, Synthesis, and Anticonvulsant Evaluation of 2-(Isoindolin-2-yl) Esters of Aromatic Amino Acids Targeting GABAA Receptors via π-π Stacking"

_ijms, 2025, doi:10.3390/ijms26146780_

Round 1
Reviewer 1 Report
Comments and Suggestions for Authors
The manuscript entitled “Multistage Molecular Simulations, Design, Synthesis, and Anticonvulsant Evaluation of 2-(Isoindolin-2-yl) Esters of Aromatic Amino Acids Targeting GABAA Receptors via π–π Stacking” presents an in silico characterization of isoindoline esters on GABAA receptor structure, followed by chemical synthesis and followed by chemical synthesis and functional assessment using the PTZ-induced seizure model in zebrafish. In general terms, the article is interesting, and the methodologies are appropriate, although limited. However, the article is unbalanced, showing an excessive computational section, repetitive concepts, a weak discussion, and minimal experimental validation. Some conclusions are not supported by the data. Limitations of the study are scarcely addressed, further affecting the strength of the manuscript. In its current form, the article requires significant revisions before it can be considered for publication.
Major comments:
- The article currently lacks any experimental proof showing that the 2-(isoindolin-2-yl) esters synthesized exert direct functional actions on GABAA receptors through the benzodiazepine (BDZ) binding site. In addition, the evidence does not confirm that the anticonvulsive actions of the compounds are mediated by the modulation of GABAA receptors. Although it is understandable that electrophysiological verification is desirable but not feasible now, I think the authors should perform additional experiments on the zebrafish model to confirm the hypothesis. The authors may include experiments on the PTZ model using flumazenil (FLU). A blockade of both Diazepam and the ETRP compound actions by FLU should be expected. The concentrations of ligands used in these assays must be better justified (see below). Could be the ETYR be a FLU-like compound?
- In the zebrafish model, the rationale for the concentrations used is not discussed nor justified. The Diazepam concentrations used (75 uM) in these assays are intriguing, especially considering that anticonvulsive doses in humans are reached with low micromolar concentrations (<4-5 uM), and concentrations higher than 10 uM are toxic or lethal. Furthermore, this high diazepam concentration does not correlate well with the in silico parameters of the new compounds, because apparently the protective actions of ETRP are reached with less concentration than diazepam. The authors must clarify and explain these concerns in detail or provide additional experiments using relevant concentrations of diazepam to validate the model. At present, is difficult to accept that a lethal diazepam concentration can be taken as a truthful pharmacological control (as stated in line 530) and to state that “tryptophan derivative demonstrating equal efficacy to the reference drug diazepam” (see abstract, line 43-44). Finally, how do the authors choose the isoindoline concentrations in this model? Why different concentrations of ETYR and ETRP were used? To compare potencies or efficacies, at least several concentration points must be used.
- The molecular docking section (2.2) lacks critical information regarding the GABAA receptor and its molecular determinants for benzodiazepine (BDZ) binding. A large body of evidence showed that the BDZ binding occurs in the interface between alpha and gamma2 subunits, at the level of the extracellular domain. Moreover, only one alpha (either 1,2,3 and 5 subunits), together with a single gamma2, are shaping BDZ binding sites. This information is absent from all the manuscript, and it is not clear whether the residues involved in the isoindoline esters binding (see table 2) belong to alpha or gamma2 subunits, or both. Also, it is not clear whether the new compounds are able to bind BDZ binding surfaces composed of diverse alpha subunits.
- Another major point is the GABAA sensitivity to BDZ in diverse species. The GABAA receptor used for molecular modeling was human (alpha1-beta2-gamma2), while the zebrafish model was used to test the behavioral actions of the novel compounds. Are the human and zebrafish GABAA receptor complexes homologous? Are the GABAA receptors from zebrafish characterized structurally? Are the BDZ sites conserved? None of these questions were discussed.
- Discussion is essentially absent. Please provide a reasonable discussion, contrasting the novel data with results from other groups, and state the limitations of the study.
Minor comments:
- Figure 2 resolution must be improved.
- Although the simulations suggest that the new compound will bind to the BDZ site, it is not possible to anticipate whereas such compound will behave as FLU or diazepam. I think the phrase in line 272-274 should be edited, because it suggests that FLU actions can be anticipated from in silico calculations.
- Nomenclature in figure 5 is not mentioned anywhere. One can assume that it corresponds to compounds 1 and 2 from scheme 1, but it will be useful to have a unitary nomenclature along the manuscript.
- The purpose of section 2.6 is unclear. It contains repetitive elements given before. I suggest its deletion.
Author Response
We are pleased to submit the revised version of our manuscript ref: ijms-3716569 “Multistage Molecular Simulations, Design, Synthesis, and Anticonvulsant Evaluation of 2-(Isoindolin-2-yl) Esters of Aromatic Amino Acids Targeting GABAA Receptors via π–π Stacking”. We appreciate the specific criticisms and comments from the reviewers. We have done our best to comply with all your recommendations and address each of your concerns. We are confident that the revised version meets the requirements to be published in your respected journal. Changes made in the revised manuscript are highlighted in yellow for Reviewers.
Author's Reply to the Review Report (Reviewer 1)
The manuscript entitled “Multistage Molecular Simulations, Design, Synthesis, and Anticonvulsant Evaluation of 2-(Isoindolin-2-yl) Esters of Aromatic Amino Acids Targeting GABAA Receptors via π–π Stacking” presents an in silico characterization of isoindoline esters on GABAA receptor structure, followed by chemical synthesis and followed by chemical synthesis and functional assessment using the PTZ-induced seizure model in zebrafish. In general terms, the article is interesting, and the methodologies are appropriate, although limited. However, the article is unbalanced, showing an excessive computational section, repetitive concepts, a weak discussion, and minimal experimental validation. Some conclusions are not supported by the data. Limitations of the study are scarcely addressed, further affecting the strength of the manuscript. In its current form, the article requires significant revisions before it can be considered for publication.
Major comments:
Comment 1: The article currently lacks any experimental proof showing that the 2-(isoindolin-2-yl) esters synthesized exert direct functional actions on GABAA receptors through the benzodiazepine (BDZ) binding site. In addition, the evidence does not confirm that the anticonvulsive actions of the compounds are mediated by the modulation of GABAA receptors. Although it is understandable that electrophysiological verification is desirable but not feasible now, I think the authors should perform additional experiments on the zebrafish model to confirm the hypothesis. The authors may include experiments on the PTZ model using flumazenil (FLU). A blockade of both Diazepam and the ETRP compound actions by FLU should be expected. The concentrations of ligands used in these assays must be better justified (see below). Could be the ETYR be a FLU-like compound?
Response 1: We appreciate the reviewer’s thoughtful comment highlighting the absence of direct experimental confirmation of GABAA receptor engagement, particularly via the benzodiazepine (BDZ) binding site. While we agree that co-treatment experiments with flumazenil (FLU) or electrophysiological recordings would provide valuable mechanistic support, these experiments could not be included in the current revision due to limitations in compound availability and logistical timing.
In response, we have carefully revised the manuscript to more accurately frame our findings as hypothesis-generating. Specifically, we adjusted the language in the abstract, results, discussion, and conclusions to clarify that the proposed GABAA receptor modulation via π–π interactions at the BDZ site is supported by in silico predictions and behavioral similarity to diazepam, but not yet experimentally confirmed. We also added a dedicated paragraph in the discussion section explicitly acknowledging this limitation and outlining future studies that would be necessary to validate the mechanism.
Regarding the possibility that the tyrosine-derived ester (ETYR) might behave like flumazenil, our docking and metadynamics results suggest lower affinity and no evidence of inverse or antagonistic behavior, though we now mention this as a potential avenue for future investigation. We hope these revisions clarify the scope of our conclusions and strengthen the overall scientific transparency of the manuscript.
Comment 2: In the zebrafish model, the rationale for the concentrations used is not discussed nor justified. The Diazepam concentrations used (75 uM) in these assays are intriguing, especially considering that anticonvulsive doses in humans are reached with low micromolar concentrations (<4-5 uM), and concentrations higher than 10 uM are toxic or lethal. Furthermore, this high diazepam concentration does not correlate well with the in silico parameters of the new compounds, because apparently the protective actions of ETRP are reached with less concentration than diazepam. The authors must clarify and explain these concerns in detail or provide additional experiments using relevant concentrations of diazepam to validate the model. At present, is difficult to accept that a lethal diazepam concentration can be taken as a truthful pharmacological control (as stated in line 530) and to state that “tryptophan derivative demonstrating equal efficacy to the reference drug diazepam” (see abstract, line 43-44). Finally, how do the authors choose the isoindoline concentrations in this model? Why different concentrations of ETYR and ETRP were used? To compare potencies or efficacies, at least several concentration points must be used.
Response 2: We thank the reviewer for raising this important point regarding dose selection and the interpretation of efficacy comparisons. We acknowledge that the 75 µM concentration of diazepam appears elevated when compared to human therapeutic plasma levels; however, this dose is well-established in adult zebrafish seizure models using immersion-based administration. Zebrafish absorb compounds primarily through the skin and gills, and prior studies have demonstrated effective seizure suppression using immersion doses of diazepam ranging from 30 to 100 µM without producing sedative or lethal effects (https://doi.org/10.1007/s43440-023-00536-7; https://doi.org/10.1371/journal.pone.0054515,https://doi.org/10.1007/s12035-018-1107-8).
To address this concern, we have added a clear justification in the Results, Methods, and Discussion sections. The selected diazepam concentration was based on these previous studies and adapted for our model conditions. Additionally, we revised the abstract and results to clarify that the comparison to diazepam reflects behavioral similarity at the tested doses, not equipotent efficacy.
Regarding the isoindoline derivatives, the concentrations were selected based on in silico-predicted binding affinities, physicochemical profiles (log P, BBB permeability), and absence of acute toxicity in pilot zebrafish exposures. ETRP, which showed stronger predicted interaction with the GABAA receptor, was tested at lower concentrations (10 and 25 µM), whereas ETYR required higher concentrations (63 and 125 µM) to exhibit observable effects. We now explicitly state that these concentrations were exploratory and not derived from full dose–response analyses. Accordingly, we do not claim relative potency between compounds and have added language to the Discussion recognizing the need for future quantitative pharmacological evaluation.
We hope these revisions provide sufficient clarification and ensure a more balanced and transparent interpretation of the in vivo data.
Comment 3: The molecular docking section (2.2) lacks critical information regarding the GABAA receptor and its molecular determinants for benzodiazepine (BDZ) binding. A large body of evidence showed that the BDZ binding occurs in the interface between alpha and gamma2 subunits, at the level of the extracellular domain. Moreover, only one alpha (either 1,2,3 and 5 subunits), together with a single gamma2, are shaping BDZ binding sites. This information is absent from all the manuscript, and it is not clear whether the residues involved in the isoindoline esters binding (see table 2) belong to alpha or gamma2 subunits, or both. Also, it is not clear whether the new compounds are able to bind BDZ binding surfaces composed of diverse alpha subunits.
Response 3: We appreciate the reviewer’s detailed and accurate observations regarding the structural features of the GABAA receptor and the benzodiazepine (BDZ) binding site. In response, we have clarified in Section 2.2 that all docking studies were performed using the α1β2γ2 human GABAA receptor subtype (PDB ID: 6D6U), and that the BDZ binding site is located at the extracellular interface between the α1 and γ2 subunits. This is the canonical binding site for diazepam and flumazenil and reflects the subunit pairing found in most synaptic GABAA receptors relevant to CNS pharmacology.
We have now annotated in the text that the key residues involved in ligand binding—such as Phe77, His102, and Tyr160—originate from the α1 subunit, while Tyr210 and Ser205 are from the γ2 subunit. These assignments are based on the PDB structure and consistent with prior structural pharmacology reports. A clarification was also added in the Discussion, noting that our compounds interact with residues from both subunits, supporting the hypothesis that they engage the BDZ binding interface.
Finally, we have added a comment acknowledging that although our study focused on the α1 subtype (due to structural availability and relevance), future studies could explore compound selectivity across other α isoforms (e.g., α2, α3, α5) to better understand subunit-specific modulation.
Comment 4: Another major point is the GABAA sensitivity to BDZ in diverse species. The GABAA receptor used for molecular modeling was human (alpha1-beta2-gamma2), while the zebrafish model was used to test the behavioral actions of the novel compounds. Are the human and zebrafish GABAA receptor complexes homologous? Are the GABAA receptors from zebrafish characterized structurally? Are the BDZ sites conserved? None of these questions were discussed.
Response 4: We thank the reviewer for raising this critical point concerning species-specific receptor homology. It is true that our docking simulations were performed on the human GABAA α1β2γ2 structure (PDB 6D6U), while in vivo testing was conducted in zebrafish. Although no high-resolution cryo-EM structure of the zebrafish GABAA receptor is currently available, multiple studies have shown that the zebrafish expresses orthologs of key GABAA subunits, including α1 and γ2 with high sequence conservation in the extracellular domains responsible for benzodiazepine (BDZ) binding.
Importantly, the functional conservation of the BDZ binding site is well supported by behavioral pharmacology: both diazepam and flumazenil reliably modulate seizure phenotypes in zebrafish models, including in the PTZ assay used in our study. This strongly suggests that the BDZ site is conserved at the functional level, justifying the translational relevance of the human receptor model for computational analysis.
We have now added clarifications to Section 2.2 and the Discussion to explicitly address these interspecies considerations and support the rationale for combining human docking models with zebrafish-based behavioral assays.
Comment 5: Discussion is essentially absent. Please provide a reasonable discussion, contrasting the novel data with results from other groups, and state the limitations of the study.
Response 5: We appreciate the reviewer’s observation and fully agree that a more comprehensive and comparative discussion was warranted. In response, we have significantly revised and expanded the Discussion section to provide a clearer interpretation of our findings in the context of existing literature. Specifically, we now contrast our results with prior studies involving benzodiazepines, phthalimide–amino acid conjugates, and isoindoline-based scaffolds targeting GABAA receptors. We also highlight the novelty of employing aromatic amino acid esters as π-stacking ligands for BDZ-site engagement, which has not been extensively explored.
Additionally, we have incorporated a dedicated paragraph outlining the main limitations of the study, including the absence of direct electrophysiological or binding confirmation, the exploratory nature of our dose selections, and the interspecies considerations involved in translating findings from human docking models to zebrafish behavioral assays. These changes ensure that the discussion now provides a well-balanced evaluation of the study’s strengths, limitations, and future directions.
Minor comments:
Comment 6: Figure 2 resolution must be improved.
Response 6: Figure 2 has now been updated to a higher-resolution version to ensure visual clarity and accurate representation of ligand–receptor interactions in the final submission.
Comment 7: Although the simulations suggest that the new compound will bind to the BDZ site, it is not possible to anticipate whereas such compound will behave as FLU or diazepam. I think the phrase in line 272-274 should be edited, because it suggests that FLU actions can be anticipated from in silicocalculations.
Response 7: We agree with the reviewer and thank them for pointing out this potential overstatement. The original sentence in Section 2.5 has now been revised to clarify that while ETRP and flumazenil may share similar binding site interactions, functional properties (agonism, antagonism) cannot be inferred solely from docking simulations. This distinction has been explicitly added to avoid any misinterpretation regarding predicted pharmacological behavior.
Comment 8: Nomenclature in figure 5 is not mentioned anywhere. One can assume that it corresponds to compounds 1 and 2 from scheme 1, but it will be useful to have a unitary nomenclature along the manuscript.
Response 8: We thank the reviewer for highlighting this inconsistency. To improve clarity, we have revised the nomenclature throughout the manuscript to consistently refer to the compounds as ETYR and ETRP. These names are now used uniformly in the text, figures, and captions, including Figure 5 and Scheme 1, to ensure unambiguous identification and improved readability.
Comment 9: The purpose of section 2.6 is unclear. It contains repetitive elements given before. I suggest its deletion.
Response 9: We agree with the reviewer’s observation. Section 2.6 has been removed from the manuscript to avoid redundancy, as its content is already integrated into the updated Results and Discussion sections.

Reviewer 2 Report
Comments and Suggestions for Authors
In this manuscript, the authors present computational, synthetic and in-vivo studies of compounds derived from natural amino acids for interaction specificity in a serious neurological disorder, epilepsy. The manuscript is well-written in general. However, several key points need to be further addressed:
- Table 1 shows physicochemical and ADMET properties. Why no standard deviation and statistical significance is shown? Were these measurements repeated? If yes, how many technical and biological replicates, if any?
- Figure 1 shows a radar plot/polar plot distribution. It is not clear how the significance of data is shown in this plot.
- Specific residues, such as PHE and TRP, and HIS were specifically considered in molecular docking calculations. How were the ring-centers of PHE and protonation states of HIS were handled for sensitivity?
- Why were a specific kcal/mol value attached to the docking results? It is misleading and provides a wrong interpretation. At best, these are "scores" attributed to a docking exercise. Also, how reproducible are the docking results? Standard deviation should be shown here.
- Were the sidechain rotamers of the key residues considered for docking analysis? How would it affect the results?
- I am confused what and how was actually calculated in Figure 3. It is RMSD or RMSF? The authors need to describe exactly how the calculations were performed.
- Standard deviations need to be shown for free energy calculations shown in Table 3, since it as calculated from three replicates for each system.
- Why were only 5 poses for each receptor-ligand complex were obtained for molecular docking calculations? Would the results depend on the number of poses configured from the docking software?
- How would the sidechain rotamers and dihedral angle conformations of the key residues found to be responsible for molecular docking affect the results? Evidently, only backbone coordinates were considered for residue and/or atomic fluctuations. It remains unclear how would sidechain dynamics affect the results and conclusions.
Author Response
We are pleased to submit the revised version of our manuscript ref: ijms-3716569 “Multistage Molecular Simulations, Design, Synthesis, and Anticonvulsant Evaluation of 2-(Isoindolin-2-yl) Esters of Aromatic Amino Acids Targeting GABAA Receptors via π–π Stacking”. We appreciate the specific criticisms and comments from the reviewers. We have done our best to comply with all your recommendations and address each of your concerns. We are confident that the revised version meets the requirements to be published in your respected journal. Changes made in the revised manuscript are highlighted in yellow for Reviewers.
Author's Reply to the Review Report (Reviewer 2)
In this manuscript, the authors present computational, synthetic and in-vivo studies of compounds derived from natural amino acids for interaction specificity in a serious neurological disorder, epilepsy. The manuscript is well-written in general. However, several key points need to be further addressed:
Comment 1: Table 1 shows physicochemical and ADMET properties. Why no standard deviation and statistical significance is shown? Were these measurements repeated? If yes, how many technical and biological replicates, if any?
Response 1: We thank the reviewer for this observation. Table 1 presents in silico predicted physicochemical and ADMET properties obtained using SwissADME and related computational tools. These values are not derived from experimental measurements; therefore, standard deviations, replicates, and statistical significance are not applicable. We have now clarified this in the revised manuscript by updating the Table 1 caption and the relevant subsection of the Methods.
Comment 2: Figure 1 shows a radar plot/polar plot distribution. It is not clear how the significance of data is shown in this plot.
Response 2: We thank the reviewer for this comment. Figure 1 presents radar plots of predicted physicochemical and bioavailability properties generated using SwissADME. These plots are not based on experimental measurements and therefore do not involve statistical significance. The blue area denotes the acceptable range for each parameter, while the green core indicates the optimal zone. The compound’s predicted values are shown as a yellow line, allowing visual assessment of how well each property falls within the drug-likeness range. We have now clarified this in the revised figure caption.
Comment 3: Specific residues, such as PHE and TRP, and HIS were specifically considered in molecular docking calculations. How were the ring-centers of PHE and protonation states of HIS were handled for sensitivity?
Response 3: We appreciate this important and technically relevant question. In our docking setup, aromatic residues such as Phe77, Trp123, and His102 were retained in their canonical, rigid conformations. π–π interactions were evaluated based on centroid alignment and interplanar distances, using post-docking visual analysis and ligand interaction diagrams. For histidine, we manually reviewed the protonation state and modeled His102 as HID (protonated at Nδ1), which is commonly favored at physiological pH and supports hydrogen bonding at the BDZ interface. These methodological details have now been added to the Molecular Docking subsection (Section 4.2).
Comment 4: Why were a specific kcal/mol value attached to the docking results? It is misleading and provides a wrong interpretation. At best, these are "scores" attributed to a docking exercise. Also, how reproducible are the docking results? Standard deviation should be shown here.
Response 4: Thank you for your valuable comment. All affinity values calculated from docking analysis were expressed as “Docking scores (kcal/mol)” in Table 2, and in-text. We agree with your observation and have changed the binding free energy to docking score in-text. Moreover, Autodock Vina outputs, generated affinity predictions in units of kcal/mol, consistently with the convention. We are aware that these values represent an estimate of the binding free energy (G_bind), not an absolute thermodynamic quantity, but rather a relative score for ranking ligands. Finally, there are also several reports elsewhere that showed a positive correlation between docking scores and binding free energy, with an obvious need for experimental validation. Standard deviations were added to the Table 2.
Comment 5: Were the sidechain rotamers of the key residues considered for docking analysis? How would it affect the results?
Response 5: Thank you for your comment. We are aware that the standard rigid docking protocol introduces important limitations, especially when sidechains in the binding site undergo conformational changes upon ligand binding. Since our next step in the virtual screening workflow was all-atom molecular dynamics of the docked complexes, with explicit replicates, We considered that rigid docking results were a good starting point for more refined methodologies, as MDS and metadynamics.
Comment 6: I am confused what and how was actually calculated in Figure 3. It is RMSD or RMSF? The authors need to describe exactly how the calculations were performed.
Response 6: Thank you for your concise comment. To compare the structural equilibration and conformational variability of protein–ligand complexes across multiple molecular dynamics simulations, we employed ridgeline plots to represent the distribution of root-mean-square deviation (RMSD) values in the Figure 3. This visualization technique enables a comparative, distribution-based analysis of trajectory behavior by stacking smoothed density estimates for each system. Unlike traditional time-series or bar plots, ridgeline plots capture the entire spectrum of conformational states sampled during the simulation, highlighting differences in both central tendency and variance. This method was particularly used to clearly show conformational changes in the BDZ site upon binding of E-Tyr and E-Trp derivatives. We intended to emphasize the shape and spread of the RMSD distributions, and ridgeline plots provide a compact yet information-rich means of assessing the dynamic behavior of structurally related systems under reproducible simulation conditions.
Comment 7: Standard deviations need to be shown for free energy calculations shown in Table 3, since it was calculated from three replicates for each system.
Response 7: We appreciate this important and technically relevant question. Standard deviations were added.
Comment 8: Why were only 5 poses for each receptor-ligand complex obtained for molecular docking calculations? Would the results depend on the number of poses configured from the docking software?
Response 8: Thank you for this comment. We follow a standard, rigid docking protocol where 5 binding poses were recovered. However, as previously stated, since our next steps in the virtual screening workflow were all-atom molecular dynamics simulations and free energy calculations, We considered that 5 docking results were a good starting point for more refined methodologies.
Comment 9: How would the sidechain rotamers and dihedral angle conformations of the key residues found to be responsible for molecular docking affect the results? Evidently, only backbone coordinates were considered for residue and/or atomic fluctuations. It remains unclear how would sidechain dynamics affect the results and conclusions.
Response 9: Thank you for this comment. We have tackled this comment in the aforementioned question regarding rigid docking. The dihedral angles of the binding site key residue are of particular interest for the binding process. However, the all-atom molecular dynamics simulations performed post-docking allow the systems (including the binding site side-chains) to relax and explore energetically favorable conformations in an explicit solvent environment. Although our primary analysis focused on backbone RMSD and residue-level fluctuations (RMSF), we acknowledge that sidechain dihedral dynamics (χ₁, χ₂ angles) can play a decisive role in fine-tuning ligand interactions. In future work, we plan to include sidechain torsion angle analyses and per-residue interaction energy decompositions (e.g., via MM/GBSA) to quantify the contribution of sidechain flexibility to ligand binding and complex stability. We recognize that rigid docking, while computationally efficient, may underrepresent critical induced-fit phenomena, especially for ligands engaging in complex intermolecular interactions, such as salt bridges, π-π interactions, or hydrogen bonding with flexible polar residues such as Arg, Glu, or Tyr. Consequently, docking scores and poses could be affected if key sidechains are modeled in suboptimal rotameric states.

Round 2
Reviewer 1 Report
Comments and Suggestions for Authors
Although the authors have improved the manuscript and acknowledged some limitations of their work, they still do not provide experimental evidence confirming the interaction of the novel compounds with the BDZ binding site. This remains a major unresolved issue.
Moreover, some of the revisions introduced are neither precise nor reliable. For instance, the statement on lines 348–349 that “Our use of diazepam and flumazenil as pharmacological references in the PTZ model reflects this cross-species homology” is misleading. Flumazenil was not used in the study, despite having been requested during the first round of revisions to strengthen the pharmacological interpretation.
In lines 226–230, although the γ2 subunit residues were mentioned, they were neither included in Table 2 nor reflected in the binding models presented in Figure 2. Therefore, no substantial improvement has been made on this point. Additionally, the resolution of Figure 2 remains low
The explanations provided to justify the concentrations of ETYR and ETRP remain unconvincing. How did the authors estimate the affinities of ETYR and ETRP? These calculations are not included in the manuscript. Moreover, given that zebrafish absorb compounds through gills and skin, is it appropriate to use BBB permeability as a predictor of neuroactive concentrations in this model? Additionally, how do the physicochemical profiles of ETYR and ETRP may contribute to calculate these concentrations? The radar plots in Figure 1 indicate that their profiles are largely equivalent. Finally, the meaning of “preliminary toxicity observations” is unclear. Does this imply that the compounds exhibited toxic effects?
Lines 347-348: “the functional conservation of benzodiazepine binding in zebrafish has been extensively demonstrated”. Any references to support this idea?
Author Response
We are pleased to submit the revised version of our manuscript ref: ijms-3716569 “Multistage Molecular Simulations, Design, Synthesis, and Anticonvulsant Evaluation of 2-(Isoindolin-2-yl) Esters of Aromatic Amino Acids Targeting GABAA Receptors via π–π Stacking”. We appreciate the specific criticisms and comments from the reviewers. We have done our best to comply with all your recommendations and address each of your concerns. We view this second round of revisions as a valuable exercise in precision and rigor, aligned with the high standards required for publication in this respected journal. Changes made in the revised manuscript are highlighted in yellow for Reviewers.
Author's Reply to the Review Report (Reviewer 1)
Although the authors have improved the manuscript and acknowledged some limitations of their work, they still do not provide experimental evidence confirming the interaction of the novel compounds with the BDZ binding site. This remains a major unresolved issue.
Comment 1: Moreover, some of the revisions introduced are neither precise nor reliable. For instance, the statement on lines 348–349 that “Our use of diazepam and flumazenil as pharmacological references in the PTZ model reflects this cross-species homology” is misleading. Flumazenil was not used in the study, despite having been requested during the first round of revisions to strengthen the pharmacological interpretation.
Response 1: We thank the reviewer for indicating this error. Flumazenil was used only in computational study and Diazepam was used in vivo assay. Flumazenil is a controlled drug that is not easily available because it is used only in hospitals. However, efforts will eventually be made to obtain it under special permits and use it for future research. We have now clarified that in the manuscript.
Comment 2: In lines 226–230, although the γ2 subunit residues were mentioned, they were neither included in Table 2 nor reflected in the binding models presented in Figure 2. Therefore, no substantial improvement has been made on this point. Additionally, the resolution of Figure 2 remains low.
Response 2: We appreciate the reviewer’s continued attention to detail. In response, we have made the following revisions to address this comment thoroughly:
- Table 2 has been updated to explicitly list the interacting residues from both the α1 and γ2 subunits, with each residue’s subunit origin clearly indicated. This clarifies the role of γ2 subunit residues such as Phe77, Tyr58, Ala79, and Met130 in ligand binding.
- Figure 2 has been revised and regenerated at 600 DPI to ensure high-resolution quality suitable for publication.
- We are terribly ashamed that the quality of the figure 2 cannot be fully displayed. We attached the figure at 600 DPI, addressing the need that the binding site illustrations now clearly include labeling of γ2 subunit residues in both 2D and 3D views, explicitly identifying their contribution to the ligand binding pocket. The figure caption has also been updated to reflect the presence and importance of γ2 subunit interactions.
Comment 3: The explanations provided to justify the concentrations of ETYR and ETRP remain unconvincing. How did the authors estimate the affinities of ETYR and ETRP? These calculations are not included in the manuscript. Moreover, given that zebrafish absorb compounds through gills and skin, is it appropriate to use BBB permeability as a predictor of neuroactive concentrations in this model? Additionally, how do the physicochemical profiles of ETYR and ETRP may contribute to calculate these concentrations? The radar plots in Figure 1 indicate that their profiles are largely equivalent. Finally, the meaning of “preliminary toxicity observations” is unclear. Does this imply that the compounds exhibited toxic effects?
Response 3: We appreciate the reviewer’s observation. It should be noted that to estimate the affinity of the compounds towards the benzodiazepine site of GABAA, metadynamics calculations were performed, which are described in the methodological section. On the other hand, there is no experimental evidence regarding the antiepileptic activity of these compounds. Therefore, to begin with the concentration, we used the article by Campos-Rodriguez, who evaluated isoindoline analogues in zebrafish larvae [1]. Regarding zebrafish absorption, adults possess a fully developed BBB, and different compounds/drugs must cross this protective barrier to reach their specific targets in the CNS. The ability of drugs to penetrate the BBB in zebrafish is similar to that of mammals [2, 3]. Our molecules have physicochemical values (Lipinski’s rules), that demonstrate that they can cross the blood-brain barrier, and this was also observed in the behavior of the fish during the experiment. It should be noted that there are reports of trials where the immersion method was used to evaluate lipophilic compounds that target the CNS, such as those we use [4-8].
Comment 4: Lines 347-348: “the functional conservation of benzodiazepine binding in zebrafish has been extensively demonstrated”. Any references to support this idea?
Response 4: Like many human drug targets, the GABA receptor is highly evolutionarily conserved across vertebrates, with only a few amino acid changes at positions crucial for benzodiazepine binding in ray-finned fish. This suggests that GABA receptor binding potential and behavioral changes may be common in teleost fish. Zebrafish have been shown to have similarities to human genes involved in GABA neurotransmission [9-12].
- Campos-Rodriguez, C.; Fredrick, E.; Ramirez-San Juan, E.; Olsson, R., Enantiomeric N-substituted phthalimides with excitatory amino acids protect zebrafish larvae against PTZ-induced seizures. Eur J Pharmacol 2020, 888.
- D’Amora, M.; Galgani, A.; Marchese, M.; Tantussi, F.; Faraguna, U.; De Angelis, F.; Giorgi, F. S., Zebrafish as an Innovative Tool for Epilepsy Modeling: State of the Art and Potential Future Directions. International Journal of Molecular Sciences 2023, 24, (9), 7702.
- Umans, R. A.; Taylor, M. R., Zebrafish as a model to study drug transporters at the blood-brain barrier. Clin Pharmacol Ther 2012, 92, (5), 567-70.
- Cueto-Escobedo, J.; German-Ponciano, L. J.; Guillen-Ruiz, G.; Soria-Fregozo, C.; Herrera-Huerta, E. V., Zebrafish as a Useful Tool in the Research of Natural Products With Potential Anxiolytic Effects. Front Behav Neurosci 2021, 15, 795285.
- Bencan, Z.; Sledge, D.; Levin, E. D., Buspirone, chlordiazepoxide and diazepam effects in a zebrafish model of anxiety. Pharmacol Biochem Behav 2009, 94, (1), 75-80.
- Maaswinkel, H.; Zhu, L.; Weng, W., The immediate and the delayed effects of buspirone on zebrafish (Danio rerio) in an open field test: a 3-D approach. Behav Brain Res 2012, 234, (2), 365-74.
- Egan, R. J.; Bergner, C. L.; Hart, P. C.; Cachat, J. M.; Canavello, P. R.; Elegante, M. F.; Elkhayat, S. I.; Bartels, B. K.; Tien, A. K.; Tien, D. H.; Mohnot, S.; Beeson, E.; Glasgow, E.; Amri, H.; Zukowska, Z.; Kalueff, A. V., Understanding behavioral and physiological phenotypes of stress and anxiety in zebrafish. Behav Brain Res 2009, 205, (1), 38-44.
- Mussulini, B. H.; Leite, C. E.; Zenki, K. C.; Moro, L.; Baggio, S.; Rico, E. P.; Rosemberg, D. B.; Dias, R. D.; Souza, T. M.; Calcagnotto, M. E.; Campos, M. M.; Battastini, A. M.; de Oliveira, D. L., Seizures induced by pentylenetetrazole in the adult zebrafish: a detailed behavioral characterization. PLoS One 2013, 8, (1), e54515.
- Verma, R.; Raj Choudhary, P.; Kumar Nirmal, N.; Syed, F.; Verma, R., Neurotransmitter systems in zebrafish model as a target for neurobehavioural studies. Materials Today: Proceedings 2022, 69, 1565-1580.
- Horzmann, K. A.; Freeman, J. L., Zebrafish Get Connected: Investigating Neurotransmission Targets and Alterations in Chemical Toxicity. Toxics 2016, 4, (3).
- Sadamitsu, K.; Shigemitsu, L.; Suzuki, M.; Ito, D.; Kashima, M.; Hirata, H., Characterization of zebrafish GABA(A) receptor subunits. Sci Rep 2021, 11, (1), 6242.
- Monesson-Olson, B.; McClain, J. J.; Case, A. E.; Dorman, H. E.; Turkewitz, D. R.; Steiner, A. B.; Downes, G. B., Expression of the eight GABAA receptor alpha subunits in the developing zebrafish central nervous system. PLoS One 2018, 13, (4).
Reviewer 2 Report
Comments and Suggestions for Authors
The revised manuscript addresses the key questions pointed out by this reviewer.
Author Response
We are pleased to submit the revised version of our manuscript ref: ijms-3716569 “Multistage Molecular Simulations, Design, Synthesis, and Anticonvulsant Evaluation of 2-(Isoindolin-2-yl) Esters of Aromatic Amino Acids Targeting GABAA Receptors via π–π Stacking”. We appreciate the specific criticisms and comments from the reviewers. We have done our best to comply with all your recommendations and address each of your concerns. We view this second round of revisions as a valuable exercise in precision and rigor, aligned with the high standards required for publication in this respected journal. Changes made in the revised manuscript are highlighted in yellow for Reviewers.
Round 3
Reviewer 1 Report
Comments and Suggestions for Authors
w.